# Hyperbolic Discounting and Learning over Multiple Horizons

## Abstract

Reinforcement learning (RL) typically defines a discount factor ($\gamma$) as part of the Markov Decision Process. The discount factor values future rewards by an exponential scheme that leads to theoretical convergence guarantees of the Bellman equation. However, evidence from psychology, economics and neuroscience suggests that humans and animals instead have *hyperbolic* time-preferences ($\frac{1}{1+kt}$ for $k > 0$). Here we extend earlier work of Kurth-Nelson and Redish and propose an efficient deep reinforcement learning agent that acts via hyperbolic discounting and other non-exponential discount mechanisms. We demonstrate that a simple approach approximates hyperbolic discount functions while still using familiar temporal-difference learning techniques in RL. Additionally, and independent of hyperbolic discounting, we make a surprising discovery that simultaneously learning value functions over multiple time-horizons is an effective auxiliary task which often improves over state-of-the-art methods.

## 1 Introduction

The standard treatment of the reinforcement learning (RL) problem is the Markov Decision Process (MDP) which includes a discount factor $0 \leq \gamma \leq 1$ that exponentially reduces the present value of future rewards (Bellman, 1957; Sutton & Barto, 1998). A reward $r_t$ received in $t$-time steps is devalued to $\gamma^t r_t$, a discounted utility model introduced by Samuelson (1937). This establishes a time-preference for rewards realized sooner rather than later. The decision to exponentially discount future rewards by $\gamma$ leads to value functions that satisfy theoretical convergence properties (Bertsekas, 1995). The magnitude of $\gamma$ also plays a role in stabilizing learning dynamics of RL algorithms (Prokhorov & Wunsch, 1997; Bertsekas & Tsitsiklis, 1996) and has recently been treated as a hyperparameter of the optimization (OpenAI, 2018; Xu et al., 2018).

However, both the magnitude and the functional form of this discounting function establish priors over the solutions learned. The magnitude of $\gamma$ chosen establishes an *effective horizon* for the agent of $1/(1-\gamma)$, far beyond which rewards are neglected (Kearns & Singh, 2002). This effectively imposes a time-scale of the environment, which may not be accurate. Further, the exponential discounting of future rewards is consistent with a prior belief that there is a known constant per-time-step hazard rate (Sozou, 1998) or probability of dying of $1 - \gamma$ (Lattimore & Hutter, 2011).

Additionally, discounting future values exponentially and according to a single discount factor $\gamma$ does not harmonize with the measured value preferences in humans[1] and animals (Mazur, 1985; 1997; Ainslie, 1992; Green & Myerson, 2004; Maia, 2009). A wealth of empirical evidence has been amassed that humans, monkeys, rats and pigeons instead discount future returns *hyperbolically*, where $d_k(t) = \frac{1}{1+kt}$, for some positive $k > 0$ (Ainslie, 1975; 1992; Mazur, 1985; 1997; Frederick et al., 2002; Green et al., 1981; Green & Myerson, 2004).

This discrepancy between the time-preferences of animals from the exponential discounted measure of value might be presumed irrational. But Sozou (1998) showed that hyperbolic time-preferences is mathematically consistent with the agent maintaining some uncertainty over the prior belief of the *hazard rate* in the environment. Hazard rate $h(t)$ measures the per-time-step risk the agent incurs as it acts in the environment due to a potential early death. Precisely, if $s(t)$ is the probability that the

---

[1]Time-preference reversals are one implication. Consider two hypothetical choices: (1) a stranger offers $1M now or $1.1M dollars tomorrow (2) a stranger instead offers $1M in 99 days versus $1.1M in 100 days.

agent is alive at time $t$ then the hazard rate is $h(t) = -\frac{d}{dt}\ln s(t)$. We consider the case where there is a fixed, but potentially unknown hazard rate $h(t) = \lambda \geq 0$. The prior belief of the hazard rate $p(\lambda)$ implies a specific discount function Sozou (1998). Under this formalism, the canonical case in RL of discounting future rewards according to $d(t) = \gamma^t$ is consistent with the belief that there exists a single hazard rate $\lambda = e^{-\gamma}$ known with certainty. Further details are available in Appendix A.

Common RL environments are also characterized by risk, but often in a narrower sense. In deterministic environments like the original Arcade Learning Environment (ALE) (Bellemare et al., 2013) stochasticity is often introduced through techniques like no-ops (Mnih et al., 2015) and sticky actions (Machado et al., 2018) where the action execution is noisy. Physics simulators may have noise and the randomness of the policy itself induces risk. But even with these stochastic injections the risk to reward emerges in a more restricted sense. In Section 2 we show that a prior distribution reflecting the uncertainty over the hazard rate, has an associated discount function in the sense that an MDP with either this hazard distribution or the discount function, has the same value function for all policies. This equivalence implies that learning policies with a discount function can be interpreted as making them robust to the associated hazard distribution. Thus, discounting serves as a tool to ensure that policies deployed in the real world perform well even under risks they were not trained under.

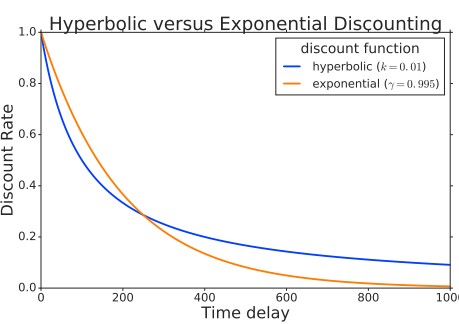

Figure 1: Hyperbolic versus exponential discounting. Humans and animals often exhibit hyperbolic discounts (blue curve) which have shallower discount declines for large horizons. In contrast, RL agents often optimize exponential discounts (orange curve) which drop at a constant rate regardless of how distant the return.

We propose an algorithm that approximates hyperbolic discounting while building on successful Q-learning (Watkins & Dayan, 1992) tools and their associated theoretical guarantees. We show learning many Q-values, each discounting exponentially with a different discount factor $\gamma$, can be aggregated to approximate hyperbolic (and other non-exponential) discount factors. We demonstrate the efficacy of our approximation scheme in our proposed Pathworld environment which is characterized both by an uncertain per-time-step risk to the agent. Conceptually, Pathworld emulates a foraging environment where an agent must balance easily realizable, small meals versus more distant, fruitful meals. We then consider higher-dimensional deep RL agents in the ALE, where we measure the benefits of hyperbolic discounting. This approximation mirrors the work of Kurth-Nelson & Redish (2009); Redish & Kurth-Nelson (2010) which empirically demonstrates that modeling a finite set of $\mu$Agents simultaneously can approximate hyperbolic discounting function. Our method then generalizes to other non-hyperbolic discount functions and uses deep neural networks to model the different Q-values from a shared representation.

Surprisingly and in addition to enabling new non-exponential discounting schemes, we observe that learning a set of Q-values is beneficial as an auxiliary task (Jaderberg et al., 2016). Adding this *multi-horizon auxiliary task* often improves over a state-of-the-art baseline, Rainbow (Hessel et al., 2018) in the ALE (Bellemare et al., 2013). This work questions the RL paradigm of learning policies through a single discount function which exponentially discounts future rewards through the following contributions:

1. **Hazardous MDPs.** We formulate MDPs with hazard present and demonstrate an equivalence between undiscounted values learned under hazards and (potentially non-exponentially) discounted values without hazard.

2. **Hyperbolic (and other non-exponential)-agent.** A practical approach for training an agent which discounts future rewards by a hyperbolic (or other non-exponential) discount function and acts according to this.

3. **Multi-horizon auxiliary task.** A demonstration of multi-horizon learning over many $\gamma$ simultaneously as an effective auxiliary task.

## 2 HAZARD IN MDPS

To study MDPs with *hazard distributions* and *general discount functions* we introduce two modifications. The hazardous MDP now is defined by the tuple $< \mathcal{S}, \mathcal{A}, R, P, \mathcal{H}, d >$. In standard form, the state space $\mathcal{S}$ and the action space $\mathcal{A}$ may be discrete or continuous. The learner observes samples from the environment transition probability $P(s_{t+1}|s_t, a_t)$ for going from $s_t \in \mathcal{S}$ to $s_{t+1} \in \mathcal{S}$ given $a_t \in \mathcal{A}$. We will consider the case where $P$ is a sub-stochastic transition function, which defines an episodic MDP. The environment emits a bounded reward $r : \mathcal{S} \times \mathcal{A} \to [r_{min}, r_{max}]$ on each transition. In this work we consider non-infinite episodic MDPs.

The first difference is that at the beginning of each episode, a hazard $\lambda \in [0, \infty)$ is sampled from the hazard distribution $\mathcal{H}$. This is equivalent to sampling a *continuing* probability $\gamma = e^{-\lambda}$. During the episode, the hazard modified transition function will be $P_\lambda$, in that $P_\lambda(s'|s, a) = e^{-\lambda} P(s'|s, a)$. The second difference is that we now consider a general discount function $d(t)$. This differs from the standard approach of exponential discounting in RL with $\gamma$ according to $d(t) = \gamma^t$, which is a special case. This setting makes a close connection to partially observable Markov Decision Process (POMDP) (Kaelbling et al., 1998) where one might consider $\lambda$ as an unobserved variable. However, the classic POMDP definition contains an explicit discount function $\gamma$ as part of its definition which does not appear here.

A policy $\pi : \mathcal{S} \to \mathcal{A}$ is a mapping from states to actions. The state action value function $Q_\pi^{\mathcal{H},d}(s, a)$ is the expected discounted rewards after taking action $a$ in state $s$ and then following policy $\pi$ until termination.

$$Q_\pi^{\mathcal{H},d}(s, a) = \mathbb{E}_\lambda \mathbb{E}_{\pi, P_\lambda} \left[ \sum_{t=0}^{\infty} d(t) R(s_t, a_t) | s_0 = s, a_0 = a \right] \tag{1}$$

where $\lambda \sim \mathcal{H}$ and $\mathbb{E}_{\pi, P_\lambda}$ implies that $s_{t+1} \sim P_\lambda(\cdot|s_t, a_t)$ and $a_t \sim \pi(\cdot|s_t)$.

### 2.1 EQUIVALENCE BETWEEN HAZARD AND DISCOUNTING

In the hazardous MDP setting we observe the same connections between hazard and discount functions delineated in Appendix A. This expresses an equivalence between the value function of an MDP with a discount and MDP with a hazard distribution.

For example, there exists an equivalence between the exponential discount function $d(t) = \gamma^t$ to the *undiscounted* case where the agent is subject to a $(1 - \gamma)$ per time-step of dying (Lattimore & Hutter, 2011). The typical Q-value (left side of Equation 2) is when the agent acts in an environment without hazard $\lambda = 0$ or $\mathcal{H} = \delta(0)$ and discounts future rewards according to $d(t) = \gamma^t = e^{-\lambda t}$ which we denote as $Q_\pi^{\delta(0),\gamma^t}(s, a)$. The alternative Q-value (right side of Equation 2) is when the agent acts under hazard rate $\lambda = -\ln \gamma$ but does not discount future rewards which we denote as $Q_\pi^{\delta(-\ln \gamma),1}(s, a)$.

$$Q_\pi^{\delta(0),\gamma^t}(s, a) = Q_\pi^{\delta(-\ln \gamma),1}(s, a) \; \forall \, \pi, s, a. \tag{2}$$

where $\delta(x)$ denotes the Dirac delta distribution at $x$. This follows from $P_\lambda(s'|s, a) = e^{-\lambda} P(s'|s, a)$

$$\mathbb{E}_{\pi, P} \left[ \sum_{t=0}^{\infty} \gamma^t R(s_t, a_t) | s_0 = s, a_0 = a \right] = \mathbb{E}_{\pi, P} \left[ \sum_{t=0}^{\infty} e^{-\lambda t} R(s_t, a_t) | s_0 = s, a_0 = a \right]$$

$$= \mathbb{E}_{\pi, P_\lambda} \left[ \sum_{t=0}^{\infty} R(s_t, a_t) | s_0 = s, a_0 = a \right]$$

We also show a similar equivalence between hyperbolic discounting and the specific hazard distribution $p_k(\lambda) = \frac{1}{k} \exp(-\lambda/k)$, where again, $\lambda \in [0, \infty)$ in Appendix E.

$$Q_\pi^{\delta(0),\Gamma_k}(s, a) = Q_\pi^{p_k,1}(s, a)$$

For notational brevity later in the paper, we will omit the explicit hazard distribution $\mathcal{H}$-superscript if the environment is not hazardous. This formulation builds upon Sozou (1998)'s relate of hazard rate and discount functions and shows that this holds for generalized Q-values in reinforcement learning.

## 3 COMPUTING NON-EXPONENTIAL Q-VALUES

We now show how one can re-purpose exponentially-discounted Q-values to compute hyperbolic (and other-non-exponential) discounted Q-values. The central challenge with using non-exponential discount strategies is that most RL algorithms use some form of TD learning (Sutton, 1988). This family of algorithms exploits the Bellman equation (Bellman, 1958) which, when using exponential discounting, relates the value function at one state with the value at the following state.

$$Q_\pi^{\gamma^t}(s,a) = \mathbb{E}_{\pi,P}[R(s,a) + \gamma Q_\pi(s',a')] \tag{3}$$

where expectation $\mathbb{E}_{\pi,P}$ denotes sampling $a \sim \pi(\cdot|s)$, $s' \sim P(\cdot|s,a)$, and $a' \sim \pi(\cdot|s')$. Being able to reuse TD methods without being constrained to exponential discounting is thus an important challenge. We propose here a scheme to deduce hyperbolic as well as other non-exponentially discounted $Q$-values when our discount function has a particular form.

**Lemma 3.1.** *Let $Q_\pi^{\mathcal{H},\gamma}(s,a)$ be the state action value function under exponential discounting in a hazardous MDP $< \mathcal{S}, \mathcal{A}, R, P, \mathcal{H}, \gamma^t >$ and let $Q_\pi^{\mathcal{H},d}(s,a)$ refer to the value function in the same MDP except for new discounting $< \mathcal{S}, \mathcal{A}, R, P, \mathcal{H}, d >$. If there exists a function $w : [0,1] \to \mathbb{R}$ such that*

$$d(t) = \int_0^1 w(\gamma)\gamma^t d\gamma \tag{4}$$

*which we will refer to as the exponential weighting condition, then*

$$Q_\pi^{\mathcal{H},d}(s,a) = \int_0^1 w(\gamma)Q_\pi^{\mathcal{H},\gamma}(s,a)d\gamma \tag{5}$$

*Proof.* Applying the condition on $d$,

$$Q_\pi^{\mathcal{H},d}(s,a) = \mathbb{E}_\lambda \mathbb{E}_{\pi,P_\lambda}\left[\sum_{t=0}^\infty \left(\int_0^1 w(\gamma)\gamma^t d\gamma\right) R(s_t,a_t)|s_0 = s, a_0 = a\right] \tag{6}$$

$$= \int_0^1 \mathbb{E}_\lambda \mathbb{E}_{\pi,P_\lambda} w(\gamma) \left[\sum_{t=0}^\infty \gamma^t R(s_t,a_t)|s_0 = s, a_0 = a\right] d\gamma \tag{7}$$

$$= \int_0^1 w(\gamma)Q_\pi^{\mathcal{H},\gamma}(s,a)d\gamma \tag{8}$$

$\square$

The exchange in the above proof is valid if $\sum_{t=0}^\infty \gamma^t R(s_t,a_t) < \infty$. The exponential weighting condition is satisfied for hyperbolic discounting and other discounting that we might want to consider (see Appendix F for examples). As an example, the hyperbolic discount can be expressed as the integral of a function $f(\gamma,t)$ for $\gamma = [0,1)$ in Equation 9.

$$\frac{1}{k}\int_{\gamma=0}^1 \gamma^{1/k+t-1}d\gamma = \frac{1}{1+kt} \tag{9}$$

This equationn tells us an integral over a function $f(\gamma,t) = \frac{1}{k}\gamma^{1/k+t-1} = w(\gamma)\gamma^t$ yields the desired hyperbolic discount factor $\Gamma_k(t) = \frac{1}{1+kt}$. This integral can be derived by Sozou's Laplace transform of the hazard rate prior $\mathcal{H} = p(\lambda)$ in Equation 18 and then applying our change of variables $\gamma = e^{-\lambda}$ relating RL discount factors to hazard rates. The computation of hyperbolic and other discount functions is demonstrated in detail in Appendix F.

This prescription gives us a tool to produce general forms of *non-exponentially* discounted Q-values using our familiar exponentially discounted Q-values traditionally learned in RL (Sutton, 1988; Sutton & Barto, 1998).

## 4 APPROXIMATING HYPERBOLIC $Q$-VALUES

Section 3 describes an equivalence between hyperbolically-discounted Q-values and integrals of exponentially-discounted Q-values, however, the method required evaluating an *infinite* set of value functions. We therefore present a practical approach to approximate discounting $\Gamma(t) = \frac{1}{1+kt}$ using a finite set of functions learned via standard $Q$-learning (Watkins & Dayan, 1992). To avoid estimating an infinite number of $Q_\pi^\gamma$-values we introduce a free hyperparameter ($n_\gamma$) which is the total number of $Q_\pi^\gamma$-values to consider, each with their own $\gamma$. We use a practically-minded approach to choose $\mathcal{G}$ that emphasizes evaluating larger values of $\gamma$ rather than uniformly choosing points and empirically performs well as seen in Section 5.

$$\mathcal{G} = [\gamma_0, \gamma_1, \cdots, \gamma_{n_\gamma}] \tag{10}$$

Our approach is described in Appendix G. Each $Q_\pi^{\gamma_i}$ computes the discounted sum of returns according to that specific discount factor $Q_\pi^{\gamma_i}(s, a) = \mathbb{E}_\pi \left[ \sum_t (\gamma_i)^t r_t | s_0 = s, a_0 = a \right]$. We previously proposed two equivalent approaches for computing hyperbolic Q-values, but for simplicity we consider the one presented in Lemma 3.1. The set of $Q$-values permits us to estimate the integral through a Riemann sum (Equation 11) which is described in further detail in Appendix I.

$$Q_\pi^\Gamma(s, a) = \int_0^1 w(\gamma) Q_\pi^\gamma(s, a) d\gamma \tag{11}$$

$$\approx \sum_{\gamma_i \in \mathcal{G}} (\gamma_{i+1} - \gamma_i) \, w(\gamma_i) \, Q_\pi^{\gamma_i}(s, a) \tag{12}$$

where we estimate the integral through a lower bound. We consolidate this entire process in Figure 11 where we show the full process of rewriting the hyperbolic discount rate, hyperbolically-discounted Q-value, the approximation and the instantiated agent. This approach is similar to that of Kurth-Nelson & Redish (2009) where each $\mu$Agent models a specific discount factor $\gamma$. However, this differs in that our final agent computes a weighted average over each Q-value rather than a sampling operation of each agent based on a $\gamma$-distribution.

## 5 HYPERBOLIC RESULTS

### 5.1 WHEN TO DISCOUNT HYPERBOLICALLY?

The benefits of hyperbolic discounting will be greatest under two conditions: uncertain hazard and non-trivial intertemporal decisions. The first condition can arise under a unobserved hazard-rate variable $\lambda$ drawn independently at the beginning of each episode from $\mathcal{H} = p(\lambda)$. The second condition emerges with a choice between a smaller nearby rewards versus larger distant rewards.[2] In the absence of both properties we would not expect any advantage to discounting hyperbolically. To see why, if there is a single-true hazard rate $\lambda_\text{env}$, than an optimal $\gamma^* = e^{-\lambda_\text{env}}$ exists and future rewards should be discounted exponentially according to it. Further, if there is a single path through the environment with perfect alignment of short- and long-term objectives, all discounting schemes yield the same optimal policy.

### 5.2 PATHWORLD EXPERIMENTS

We note two sources for discounting rewards in the future: *time delay* and *survival probability* (Section 2). In Pathworld we train to maximize hyperbolically discounted returns ($\sum_t \Gamma_k(t) R(s_t, a_t)$) under no hazard ($\mathcal{H} = \delta(\lambda - 0)$) but then evaluate the undiscounted returns $d(t) = 1.0 \ \forall \ t$ with the paths subject to hazard $\mathcal{H} = \frac{1}{k} \exp(-\lambda/k)$.

Through this procedure, we are able to train an agent that is *robust* to hazards in the environment. The agent makes one decision in Pathworld (Figure 2): which of the $N$ paths to investigate. Once a path is chosen, the agent continues until it reaches the end or until it dies. This is similar to a multi-armed bandit, with each action subject to dynamic risk. The paths vary quadratically in length with the index $d(i) = i^2$ but the rewards increase linearly with the path index $r(i) = i$. This presents

---

[2]A *trivial* intertemporal decision is one between small distant rewards versus large close rewards. For example, the choice between $100 now versus $10 tomorrow.

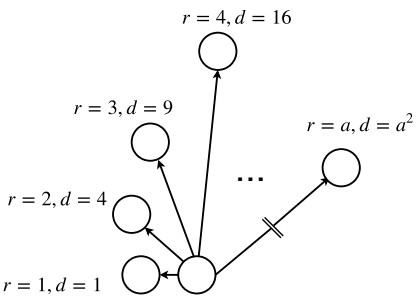

Figure 2: The Pathworld. Each state (white circle) indicates the accompanying reward $r$ and the distance from the starting state $d$. From the start state, the agent makes a single action: which which path to follow to the end. Longer paths have a larger rewards at the end, but the agent incurs a higher risk on a longer path.

a non-trivial decision for the agent. At deployment, an unobserved hazard $\lambda \sim \mathcal{H}$ is drawn and the agent is subject to a per-time-step risk of dying of $(1 - e^{-\lambda})$. This environment differs from the adjusting-delay procedure presented by Mazur (1987) and then later modified by Kurth-Nelson & Redish (2009). Rather then determining time-preferences through variable-timing of rewards, we determine time-preferences through risk to the reward.

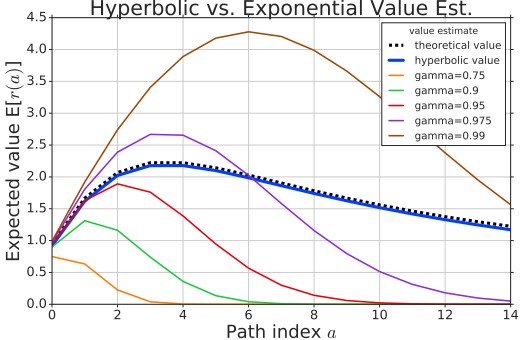

| Discount function | MSE |
|---|---|
| **hyperbolic value** | **0.002** |
| $\gamma$=0.975 | 0.566 |
| $\gamma$=0.95 | 1.461 |
| $\gamma$=0.9 | 2.253 |
| $\gamma$=0.99 | 2.288 |
| $\gamma$=0.75 | 2.809 |

Figure 3: In each episode of Pathworld an unobserved hazard $\lambda \sim p(\lambda)$ is drawn and the agent is subject to a total risk of the reward not being realized of $(1 - e^{-\lambda})^{d(a)}$ where $d(a)$ is the path length. When the agent's hazard prior matches the true hazard distribution, the value estimate agrees well with the theoretical value. Exponentially discounted values fail to approximate the true value (Table 1).

Table 1: The average mean squared error (MSE) over each of the paths in Figure 3 showing that our approximation scheme well-approximates the true value-profile.

Figure 3 validates that our approach well-approximates the true hyperbolic value of each path when the hazard prior matches the true distribution. Agents that discount exponentially according to a single $\gamma$ (the typical case in RL) incorrectly value the paths. We examine further the failure of exponential discounting in this hazardous setting. For this environment, the true hazard parameter in the prior was $k = 0.05$ (i.e. $\lambda \sim 20\exp(-\lambda/0.05)$). Therefore, at deployment, the agent must deal with dynamic levels of risk and faces a non-trivial decision of which path to follow. Even if we tune an agent's $\gamma = 0.975$ such that it chooses the correct arg-max path, it still fails to capture the functional form (Figure 3) and it achieves a high error over all paths (Table 1). If the arg-max action was not available or if the agent was proposed to evaluate non-trivial intertemporal decisions, it would act sub-optimally. In Appendix B we consider additional experiments where the agent's prior over hazard more realistically *does not* exactly match the environment true hazard rate and demonstrate the benefit of appropriate priors.

## 5.3 ATARI 2600 EXPERIMENTS

With our approach validated in Pathworld, we now move to the high-dimensional environment of Atari 2600, specifically, ALE. We use the Rainbow variant from Dopamine (Castro et al., 2018) which implements three of the six considered improvements from the original paper: distributional RL, predicting n-step returns and prioritized replay buffers. The agent (Figure 4) maintains a shared representation $h(s)$ of state, but computes $Q$-value logits for each of the $N$ $\gamma_i$ via $Q_\pi^{(i)}(s, a) = W_i h(s) + b_i$ where $W_i$ and $b_i$ are the learnable parameters of the affine transformation for that head. A ReLU-nonlinearity is used within the body of the network (Nair & Hinton, 2010).

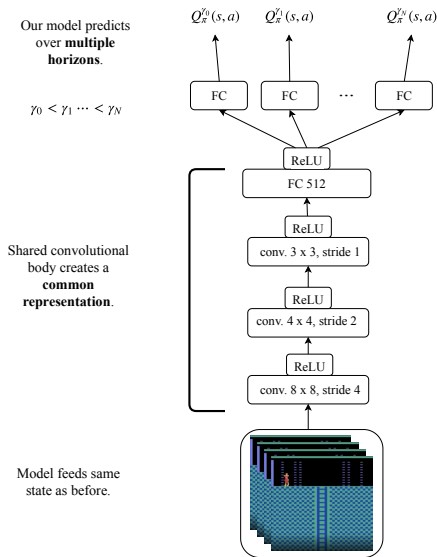

Figure 4: *Multi-horizon* model predicts $Q$-values for $n_\gamma$ separate discount functions thereby modeling different effective horizons. Each $Q$-value is a lightweight computation, an affine transformation off a shared representation. By modeling over multiple time-horizons, we now have the option to construct *policies* that act according to a particular value or a weighted combination.

Hyperparameter details are provided in Appendix K and when applicable, they default to the standard Dopamine values. We find strong performance improvements of the hyperbolic agent built on Rainbow (Hyper-Rainbow; blue bars) on a random subset of Atari 2600 games in Figure 5.

## 6 MULTI-HORIZON AUXILIARY TASK RESULTS

To dissect the Hyper-Rainbow improvements, recognize that two properties from the base Rainbow agent have changed:

1. **Behavior policy,** $\mu$**.** The agent acts according to hyperbolic Q-values computed by our approximation described in Section 4
2. **Learn over multiple horizons.** The agent simultaneously learns Q-values over many $\gamma$ rather than a Q-value for a single $\gamma$

On this subset of 19 games, Hyper-Rainbow improves upon 14 games and in some cases, by large margins. But we seek here a more complete understanding of the *underlying driver* of this improvement in ALE through an ablation study.

The second modification can be regarded as introducing an *auxiliary task* (Jaderberg et al., 2016). Therefore, to attribute the performance of each properly we construct a Rainbow agent augmented with the multi-horizon auxiliary task (referred to as Multi-Rainbow and shown in orange) but have it still act according to the original policy. That is, Multi-Rainbow acts to maximize expected rewards discounted by a fixed $\gamma_{action}$ but now learns over multiple horizons as shown in Figure 4.

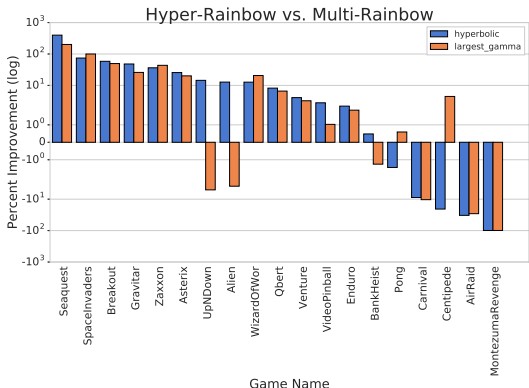

Figure 5: We compare the Hyper-Rainbow (in blue) agent versus the Multi-Rainbow (orange) agent on a random subset of 19 games from ALE (3 seeds each). For each game, the percentage performance improvement for each algorithm against Rainbow is recorded. There is no significant difference whether the agent acts according to hyperbolically-discounted (Hyper-Rainbow) or exponentially-discounted (Multi-Rainbow) Q-values suggesting the performance improvement in ALE emerges from the multi-horizon auxiliary task.

We find that the Multi-Rainbow agent performs nearly as well on these games, suggesting the effectiveness of this as a stand-alone auxiliary task. This is not entirely unexpected given the rather special-case of hazard exhibited in ALE through sticky-actions (Machado et al., 2018).

We examine further and investigate the performance of this auxiliary task across the full Arcade Learning Environment (Bellemare et al., 2017) using the recommended evaluation by (Machado et al., 2018). Doing so we find strong empirical benefits of the multi-horizon auxiliary task over the state-of-the-art Rainbow agent as shown in Figure 6.

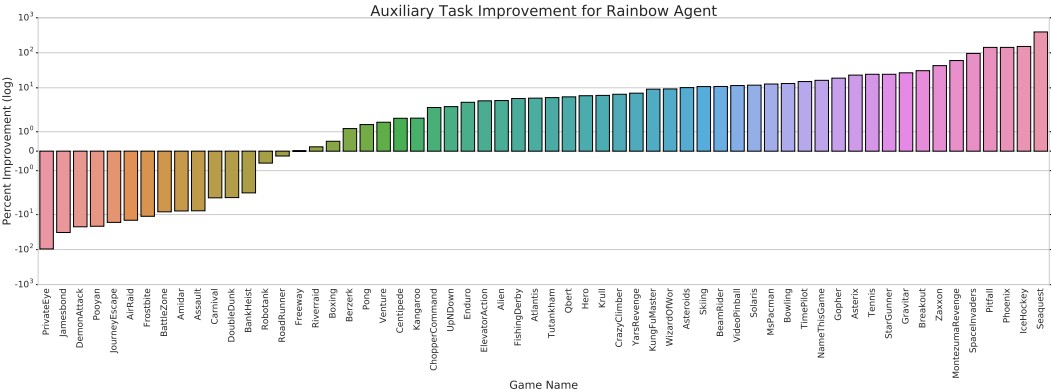

Figure 6: Performance improvement over Rainbow using the multi-horizon auxiliary task in Atari Learning Environment (3 seeds each).

### 6.1 ANALYSIS AND ABLATION STUDIES

To understand the interplay of the *multi-horizon auxiliary task* with other improvements in deep RL, we test a random subset of 10 Atari 2600 games against improvements in Rainbow (Hessel et al., 2018). On this set of games we measure a consistent improvement with multi-horizon C51 (Multi-C51) in 9 out of the 10 games over the base C51 agent (Bellemare et al., 2017) in Figure 7.

Figure 7 indicates that the current implementation of Multi-Rainbow does not generally build successfully on the prioritized replay buffer. On the subset of ten games considered, we find that four out of ten games (Pong, Venture, Gravitar and Zaxxon) are negatively impacted despite (Hessel et al., 2018) finding it to be of considerable benefit and specifically beneficial in three out of these

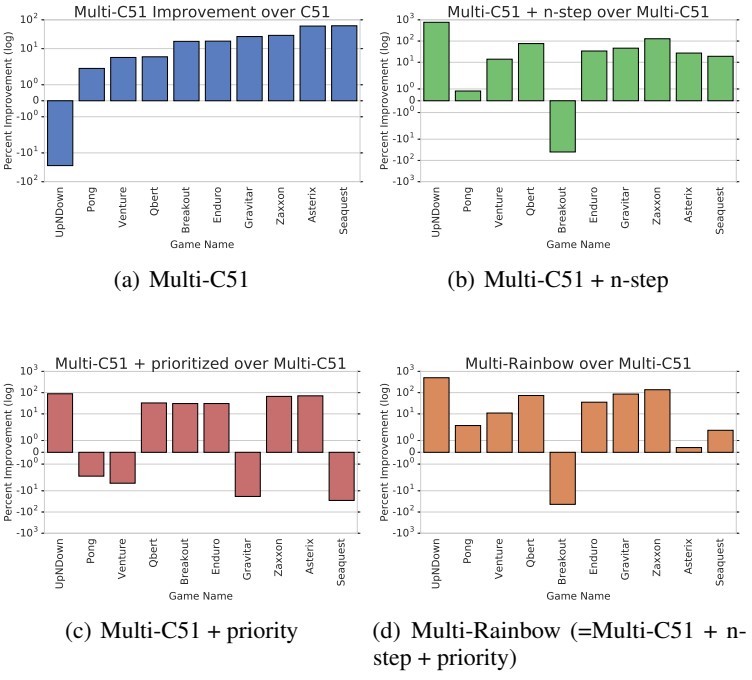

Figure 7: Measuring the Rainbow improvements on top of the Multi-C51 baseline on a subset of 10 games in the Arcade Learning Environment (3 seeds each). On this subset, we find that the multi-horizon auxiliary task interfaces well with n-step methods (top right) but poorly with a prioritized replay buffer (bottom left).

four games (Venture was not considered). The current prioritization scheme simply averaged the temporal-difference errors over all $Q$-values to establish priority. Alternative prioritization schemes are resulted in comparable performance indicating this is an open issue (Appendix J).

## 7 RELATED WORK

**Hyperbolic discounting in economics.** Hyperbolic discounting is well-studied in the field of economics (Sozou, 1998; Dasgupta & Maskin, 2005). Dasgupta and Maskin (2005) proposes a softer interpretation than Sozou (1998) (which produces a per-time-step of death via the hazard rate) and demonstrates that uncertainty over the *timing* of rewards can also give rise to hyperbolic discounting and preference reversals, a hallmark of hyperbolic discounting. Hyperbolic discounting was initially presumed to not lend itself to TD-based solutions (Daw & Touretzky, 2000) but the field has evolved on this point. Maia (2009) proposes solution directions that find models that discount quasi-hyperbolically even though each learns with exponential discounting (Loewenstein, 1996) but reaffirms the difficulty. Finally, Alexander and Brown (2010) proposes hyperbolically discounted temporal difference (HDTD) learning by making connections to hazard.

**Behavior RL and hyperbolic discounting in neuroscience.** TD-learning has long been used for modeling behavioral reinforcement learning (Montague et al., 1996; Schultz et al., 1997; Sutton & Barto, 1998). TD-learning computes the error as the difference between the expected value and actual value (Sutton & Barto, 1998; Daw, 2003) where the error signal emerges from unexpected rewards. However, these computations traditionally rely on exponential discounting as part of the estimate of the value which disagrees with empirical evidence in humans and animals (Strotz, 1955; Mazur, 1985; 1997; Ainslie, 1975; 1992). Hyperbolic discounting has been proposed as an alternative to exponential discounting though it has been debated as an accurate model (Kacelnik, 1997; Frederick et al., 2002). Naive modifications to TD-learning to discount hyperbolically present issues since the simple forms are inconsistent (Daw & Touretzky, 2000; Redish & Kurth-Nelson, 2010) RL models have been proposed to explain behavioral effects of humans and animals (Fu & Anderson, 2006;

Rangel et al., 2008) but Kurth-Nelson & Redish (2009) demonstrated that distributed exponential discount factors can directly model hyperbolic discounting. This work proposes the $\mu$Agent, an agent that models the value function with a specific discount factor $\gamma$. When the distributed set of $\mu$Agent's votes on the action, this was shown to approximate hyperbolic discounting well in the adjusting-delay assay experiments (Mazur, 1987). Using the hazard formulation established in Sozou (1998), we demonstrate how to extend this to other non-hyperbolic discount functions and demonstrate the efficacy of using a deep neural network to model the different Q-values from a shared representation.

**Towards more flexible discounting in reinforcement learning.** RL researchers have recently adopted more flexible versions beyond a fixed discount factor (Feinberg & Shwartz, 1994; Sutton, 1995; Sutton et al., 2011; White, 2017). Optimal policies are studied in Feinberg & Shwartz (1994) where two value functions with different discount factors are used. Introducing the discount factor as an argument to be queried for a set of timescales is considered in both Horde (Sutton et al., 2011) and $\gamma$-nets (Sherstan et al., 2018). Reinke et al. (2017) proposes the Average Reward Independent Gamma Ensemble framework which imitates the average return estimator. Lattimore and Hutter (2011) generalizes the original discounting model through discount functions that vary with the age of the agent, expressing time-inconsistent preferences as in hyperbolic discounting. The need to increase training stability via effective horizon was addressed in François-Lavet, Fonteneau, and Ernst (2015) who proposed dynamic strategies for the discount factor $\gamma$. Meta-learning approaches to deal with the discount factor have been proposed in Xu, van Hasselt, and Silver (2018). Finally, Pitis (2019) characterizes rational decision making in sequential processes, formalizing a process that admits a state-action dependent discount rates. Operating over multiple time scales has a long history in RL. Sutton (1995) generalizes the work of Singh (1992) and Dayan and Hinton (1993) to formalize a multi-time scale TD learning model theory. Previous work has been explored on solving MDPs with multiple reward functions and multiple discount factors though these relied on separate transition models (Feinberg & Shwartz, 1999; Dolgov & Durfee, 2005). Edwards, Littman, and Isbell (2015) considers decomposing a reward function into separate components each with its own discount factor. In our work, we continue to model the same rewards, but now model the value over different horizons. Recent work in difficult exploration games demonstrates the efficacy of two different discount factors (Burda et al., 2018) one for intrinsic rewards and one for extrinsic rewards. Finally, and concurrent with this work, Romoff et al. (2019) proposes the TD($\Delta$)-algorithm which breaks a value function into a series of value functions with smaller discount factors.

**Auxiliary tasks in reinforcement learning.** Finally, auxiliary tasks have been successfully employed and found to be of considerable benefit in RL. Suddarth and Kergosien (1990) used auxiliary tasks to facilitate representation learning. Building upon this, work in RL has consistently demonstrated benefits of auxiliary tasks to augment the low-information coming from the environment through extrinsic rewards (Lample & Chaplot, 2017; Mirowski et al., 2016; Jaderberg et al., 2016; Veeriah et al., 2018; Sutton et al., 2011)

## 8 DISCUSSION AND FUTURE WORK

This work builds on a body of work that questions one of the basic premises of RL: one should maximize the *exponentially discounted* returns via a *single* discount factor. By learning over multiple horizons simultaneously, we have broadened the scope of our learning algorithms. Through this we have shown that we can enable acting according to new discounting schemes and that learning multiple horizons is a powerful stand-alone auxiliary task. Our method well-approximates hyperbolic discounting and performs better in hazardous MDP distributions. This may be viewed as part of an algorithmic toolkit to model alternative discount functions.

However, this work still does not fully capture more general aspects of risk since the hazard rate may be a function of time. Further, hazard may not be an intrinsic property of the environment but a joint property of both the *policy* and the environment. If an agent purses a policy leading to dangerous state distributions then it will naturally be subject to higher hazards and vice-versa - this creates a complicated circular dependency. We would therefore expect an interplay between time-preferences and policy. This is not simple to deal with but recent work proposing state-action dependent discounting (Pitis, 2019) may provide a formalism for more general time-preference schemes.

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

## A  SOZOU (1998): BELIEF OF RISK IMPLIES A DISCOUNT FUNCTION

Sozou (1998) formalizes time preferences in which future rewards are discounted based on the probability that the agent will not *survive* to collect them due to an encountered risk or *hazard*.

**Definition A.1.** *Survival $s(t)$ is the probability of the agent surviving until time $t$.*

$$s(t) = P(\text{agent is alive}|\text{at time } t) \tag{13}$$

A future reward $r_t$ is less valuable presently if the agent is unlikely to survive to collect it. If the agent is risk-neutral, the present value of a future reward $r_t$ received at time-$t$ should be discounted by the probability that the agent will survive until time $t$ to collect it, $s(t)$.[3]

$$v(r_t) = s(t)r_t \tag{14}$$

Consequently, if the agent is certain to survive, $s(t) = 1$, then the reward is not discounted per Equation 14. From this it is then convenient to define the hazard rate.

**Definition A.2.** *Hazard rate $h(t)$ is the negative rate of change of the log-survival at time $t$*

$$h(t) = -\frac{d}{dt}\ln s(t) \tag{15}$$

or equivalently expressed as $h(t) = -\frac{ds(t)}{dt}\frac{1}{s(t)}$. Therefore the environment is considered hazardous at time $t$ if the log survival is decreasing sharply.

Sozou (1998) demonstrates that the prior belief of the risk in the environment implies a specific discounting function. When the risk occurs at a known constant rate than the agent should discount future rewards exponentially. However, when the agent holds *uncertainty* over the hazard rate then hyperbolic and alternative discounting rates arise.

### A.1  KNOWN HAZARD IMPLIES EXPONENTIAL DISCOUNT

We recover the familiar exponential discount function in RL based on a prior assumption that the environment has a *known constant* hazard. Consider a known hazard rate of $h(t) = \lambda \geq 0$. Definition A.2 sets a first order differential equation $\lambda = -\frac{d}{dt}\ln s(t) = -\frac{ds(t)}{dt}\frac{1}{s(t)}$. The solution for the survival rate is $s(t) = e^{-\lambda t}$ which can be related to the RL discount factor $\gamma$

$$s(t) = e^{-\lambda t} = \gamma^t \tag{16}$$

This interprets $\gamma$ as the per-time-step probability of the episode continuing. This also allows us to connect the hazard rate $\lambda \in [0, \infty]$ to the discount factor $\gamma \in [0, 1)$.

$$\gamma = e^{-\lambda} \tag{17}$$

As the hazard increases $\lambda \to \infty$, then the corresponding discount factor becomes increasingly myopic $\gamma \to 0$. Conversely, as the environment hazard vanishes, $\lambda \to 0$, the corresponding agent becomes increasingly far-sighted $\gamma \to 1$. In RL we commonly choose a single $\gamma$ which is consistent with the prior belief that there exists a known constant hazard rate $\lambda = -\ln(\gamma)$. We now relax the assumption that the agent holds this strong prior that it *exactly* knows the true hazard rate. From a Bayesian perspective, a looser prior allows for some uncertainty in the underlying hazard rate of the environment which we will see in the following section.

### A.2  UNCERTAIN HAZARD IMPLIES NON-EXPONENTIAL DISCOUNT

We may not always be so confident of the true risk in the environment and instead reflect this underlying uncertainty in the hazard rate through a hazard prior $p(\lambda)$. Our survival rate is then computed by weighting specific exponential survival rates defined by a given $\lambda$ over our prior $p(\lambda)$

$$s(t) = \int_{\lambda=0}^{\infty} p(\lambda)e^{-\lambda t}d\lambda \tag{18}$$

---

[3]Note the difference in RL where future rewards are discounted by *time-delay* so the value is $v(r_t) = \gamma^t r_t$.

Sozou (1998) shows that under an exponential prior of hazard $p(\lambda) = \frac{1}{k}\exp(-\lambda/k)$ the expected survival rate for the agent is *hyperbolic*

$$s(t) = \frac{1}{1 + kt} \equiv \Gamma_k(t) \tag{19}$$

We denote the hyperbolic discount by $\Gamma_k(t)$ to make the connection to $\gamma$ in reinforcement learning explicit. Further, Sozou (1998) shows that different priors over hazard correspond to different discount functions. We reproduce two figures in Figure 8 showing the correspondence between different hazard rate priors and the resultant discount functions. The common approach in RL is to maintain a delta-hazard (black line) which leads to exponential discounting of future rewards. Different priors lead to non-exponential discount functions.

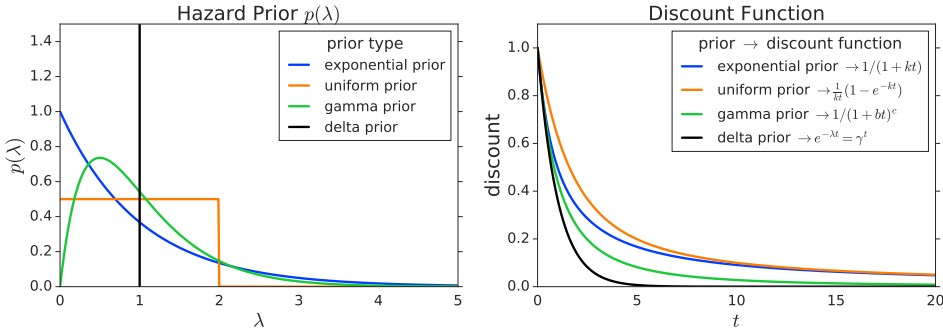

Figure 8: We reproduce two figures from Sozou (1998). There is a correspondence between hazard rate priors and the resulting discount function. In RL, we typically discount future rewards exponentially which is consistent with a Dirac delta prior (black line) on the hazard rate indicating *no uncertainty* of hazard rate. However, this is a special case and priors with uncertainty over the hazard rate imply new discount functions. All priors have the same mean hazard rate $\mathbb{E}[p(\lambda)] = 1$.

## B  ADDITIONAL PATHWORLD EXPERIMENTS

In Figure 9 we consider the case that the agent still holds an exponential prior but has the wrong coefficient $k$ and in Figure 10 we consider the case where the agent still holds an exponential prior but the true hazard is actually drawn from a uniform distribution with the same mean.

Through these two validating experiments, we demonstrate the robustness of estimating hyperbolic discounted Q-values in the case when the environment presents dynamic levels of risk and the agent faces non-trivial decisions. Hyperbolic discounting is preferable to exponential discounting even when the agent's prior does not precisely match the true environment hazard rate distribution, by coefficient (Figure 9) or by functional form (Figure 10).

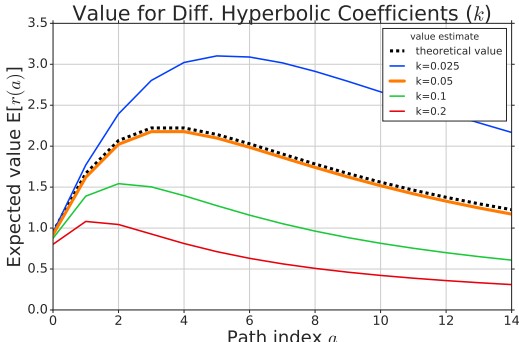

| Discount function | MSE |
|---|---|
| k=0.05 | **0.002** |
| k=0.1 | 0.493 |
| k=0.025 | 0.814 |
| k=0.2 | 1.281 |

Figure 9: Case when the hazard coefficient $k$ *does not* match that environment hazard. Here the true hazard coefficient is $k = 0.05$, but we compute values for hyperbolic agents with mismatched priors in range $k = [0.025, 0.05, 0.1, 0.2]$. Predictably, the mismatched priors result in a higher prediction error of value but performs more reliably than exponential discounting, resulting in a cumulative lower error. Numerical results in Table 2.

Table 2: The average mean squared error (MSE) over each of the paths in Figure 9. As the prior is further away from the true value of $k = 0.05$, the error increases. However, notice that the errors for large factor-of-2 changes in $k$ result in generally lower errors than if the agent had considered only a single exponential discount factor $\gamma$ as in Table 1.

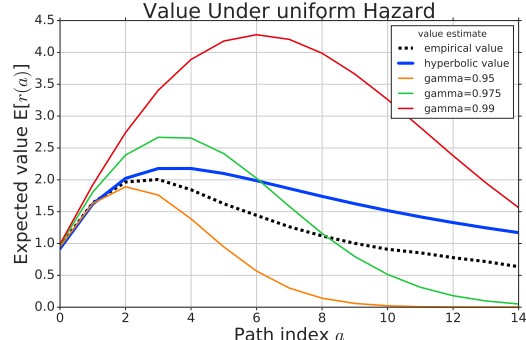

| Discount function | MSE |
|---|---|
| hyperbolic value | **0.235** |
| $\gamma = 0.975$ | 0.266 |
| $\gamma = 0.95$ | 0.470 |
| $\gamma = 0.99$ | 4.029 |

Figure 10: If the true hazard rate is now drawn according to a *uniform* distribution (with the same mean as before) the original hyperbolic discount matches the functional form better than exponential discounting. Numerical results in Table 3.

Table 3: The average mean squared error (MSE) over each of the paths in Figure 10 when the underlying hazard is drawn according to a *uniform* distribution. We find that hyperbolic discounting results is more robust to hazards drawn from a uniform distribution than exponential discounting.

## C  ALTERNATIVE APPROACH TO HYPERBOLIC Q-VALUES

### C.1  COMPUTING HYPERBOLIC $Q$-VALUES

Let's start with the case where we would like to estimate the value function where rewards are discounted hyperbolically instead of the common exponential scheme. We refer to the hyperbolic Q-values as $Q_\pi^\Gamma$ below in Equation 21

$$Q_\pi^{\Gamma_k}(s,a) = \mathbb{E}_\pi \left[ \Gamma_k(1)R(s_1,a_1) + \Gamma_k(2)R(s_2,a_2) + \cdots \middle| s,a \right] \tag{20}$$

$$= \mathbb{E}_\pi \left[ \sum_t \Gamma_k(t)R(s_t,a_t) \middle| s,a \right] \tag{21}$$

We may relate the hyperbolic $Q_\pi^\Gamma$-value to the values learned through standard $Q$-learning. To do so, notice that the hyperbolic discount $\Gamma_t$ can be expressed as the integral of a certain function $f(\gamma,t)$ for $\gamma = [0,1)$ in Equation 22.

$$\int_{\gamma=0}^{1} \gamma^{kt} d\gamma = \frac{1}{1+kt} = \Gamma_k(t) \tag{22}$$

The integral over this specific function $f(\gamma,t) = \gamma^{kt}$ yields the desired hyperbolic discount factor $\Gamma_k(t)$ by considering an *infinite set* of exponential discount factors $\gamma$ over its domain $\gamma \in [0,1)$.

Recognize that the integrand $\gamma^{kt}$ is the standard exponential discount factor which suggests a connection to standard Q-learning (Watkins & Dayan, 1992). This suggests that if we could consider an infinite set of $\gamma$ then we can combine them to yield hyperbolic discounts for the corresponding time-step $t$. We build on this idea of modeling many $\gamma$ throughout this work.

We employ Equation 22 and return to the task of computing hyperbolic Q-values $Q_\pi^\Gamma(s,a)$[4]

$$Q_\pi^\Gamma(s,a) = \mathbb{E}_\pi \left[ \sum_t \Gamma_k(t)R(s_t,a_t) \middle| s,a \right] \tag{23}$$

$$= \mathbb{E}_\pi \left[ \sum_t \left( \int_{\gamma=0}^{1} \gamma^{kt} d\gamma \right) R(s_t,a_t) \middle| s,a \right] \tag{24}$$

$$= \int_{\gamma=0}^{1} \mathbb{E}_\pi \left[ \sum_t R(s_t,a_t)(\gamma^k)^t \middle| s,a \right] d\gamma \tag{25}$$

$$= \int_{\gamma=0}^{1} Q_\pi^{(\gamma^k)^t}(s,a) d\gamma \tag{26}$$

where $\Gamma_k(t)$ has been replaced on the first line by $\left( \int_{\gamma=0}^{1} \gamma^{kt} d\gamma \right)$ and the exchange is valid if $\sum_{t=0}^{\infty} \gamma^{kt} r_t < \infty$. This shows us that we can compute the $Q_\pi^\Gamma$-value according to hyperbolic discount factor by considering an infinite set of $Q_\pi^{\gamma^k}$-values computed through standard $Q$-learning. Examining further, each $\gamma \in [0,1)$ results in TD-errors learned for a new $\gamma^k$. For values of $k < 1$, which extends the horizon of the hyperbolic discounting, this would result in larger $\gamma$.

---

[4]Hyperbolic Q-values can generally be infinite for bounded rewards. We consider non-infinite episodic MDPs only.

## D  VISUAL SUMMARY OF APPROACH

We summarize our approach for estimating non-exponential discounted Q-values here.

1. A **hyperbolic discount function**

$$\Gamma(t) = \frac{1}{1 + kt}$$

can be expressed as a **weighting over exponential discount functions** $\gamma^t$

$$\Gamma(t) = \int_{\gamma=0}^{1} w(\gamma)\ \gamma^t d\gamma$$

with weights $w(\gamma) = \frac{1}{k}\gamma^{1/k-1}$   (see *Table 1*).

2. **Hyperbolically-discounted Q-values** can be expressed as a weighting over exponentially-discounted Q-values using the same weights $w(\gamma)$:

$$Q_\pi^\Gamma(s,a) = \int_{\gamma=0}^{1} w(\gamma)\ Q_\pi^\gamma(s,a)\ d\gamma$$

3. The integral in box 2 can be **approximated with a Riemann sum** over the discrete intervals:

$$\mathcal{G} = [\gamma_0, \gamma_1 \cdots \gamma_N]$$

$$Q_\pi^\Gamma(s,a) \approx \sum_{\gamma_i \in \mathcal{G}} (\gamma_{i+1} - \gamma_i)\ w(\gamma_i)\ Q_\pi^{\gamma_i}(s,a)$$

4. Where a $Q_\pi^{\gamma_i}(s,a)$ is **simultaneously learned for each exponential discount rate** $\gamma_i \in \mathcal{G}$

$$Q_\pi^{\gamma_0}(s,a) \quad Q_\pi^{\gamma_1}(s,a) \quad \cdots \quad Q_\pi^{\gamma_N}(s,a)$$

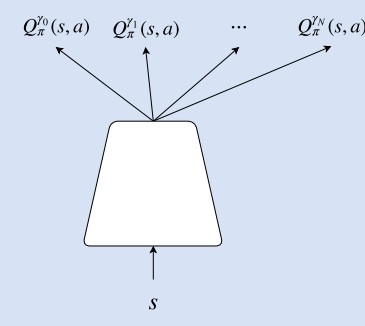

$s$

Learning simultaneous exponential Q-values are effective **auxiliary tasks.**

Figure 11: Summary of our approach to approximating hyperbolic (and other non-exponential) Q-values via a weighted sum of exponentially-discounted Q-vaulues.

# E EQUIVALENCE OF HYPERBOLIC DISCOUNTING AND EXPONENTIAL HAZARD

Following Section A we also show a similar equivalence between hyperbolic discounting and the specific hazard distribution $p_k(\lambda) = \frac{1}{k}\exp(-\lambda/k)$, where again, $\lambda \in [0,\infty)$

$$
\begin{aligned}
Q_\pi^{\delta(0),\Gamma_k}(s,a) &= \mathbb{E}_{\pi,P_0}\left[\sum_{t=0}^\infty \Gamma_k(t)R(s_t,a_t)|s_0=s,a_0=a\right] \\
&= \mathbb{E}_{\pi,P_0}\left[\sum_{t=0}^\infty \left(\int_{\lambda=0}^\infty p_k(\lambda)e^{-\lambda t}d\lambda\right)R(s_t,a_t)|s_0=s,a_0=a\right] \\
&= \int_{\lambda=0}^\infty p_k(\lambda)\mathbb{E}_{\pi,P_0}\left[\sum_{t=0}^\infty e^{-\lambda t}R(s_t,a_t)|s_0=s,a_0=a\right]d\lambda \\
&= \mathbb{E}_{\lambda\sim p_k(\cdot)}\mathbb{E}_{\pi,P_0}\left[\sum_{t=0}^\infty e^{-\lambda t}R(s_t,a_t)|s_0=s,a_0=a\right] \\
&= \mathbb{E}_{\lambda\sim p_k(\cdot)}\mathbb{E}_{\pi,P_\lambda}\left[\sum_{t=0}^\infty R(s_t,a_t)|s_0=s,a_0=a\right] \\
&= Q_\pi^{p_k,1}(s,a)
\end{aligned}
$$

Where the first step uses Equation 19. This equivalence implies that discount factors can be used to learn policies that are robust to hazards.

# F ALTERNATIVE DISCOUNT FUNCTIONS

We expand upon three special cases to see how functions $f(\gamma,t) = w(\gamma)\gamma^t$ may be related to different discount functions $d(t)$.

We summarize in Table 4 how a particular hazard prior $p(\lambda)$ can be computed via integrating over specific weightings $w(\gamma)$ and the corresponding discount function.

|  | $\mathcal{H} = p(\lambda)$ | $d(t)$ | $w(\gamma)$ |
|---|---|---|---|
| Dirac Delta Prior | $\delta(\lambda-k)$ | $e^{-kt}(=(\gamma_k)^t)$ | $\frac{1}{\gamma}\delta(-\ln\gamma-k)$ |
| Exponential Prior | $\frac{1}{k}e^{-\lambda/k}$ | $\frac{1}{1+kt}$ | $\frac{1}{k}\gamma^{1/k-1}$ |
| Uniform Prior | $\begin{cases}\frac{1}{k}, & \text{if } \lambda\in[0,k] \\ 0, & \text{otherwise}\end{cases}$ | $\frac{1}{kt}\left(1-e^{-kt}\right)$ | $\begin{cases}\frac{1}{k}\gamma^{-1}, & \text{if } \gamma\in[e^{-k},1] \\ 0, & \text{otherwise}\end{cases}$ |

Table 4: Different hazard priors $\mathcal{H} = p(\lambda)$ can be alternatively expressed through weighting exponential discount functions $\gamma^t$ by $w(\gamma)$. This table matches different hazard distributions to their associated discounting function and the weighting function per Lemma 3.1. The typical case in RL is a Dirac Delta Prior over hazard rate $\delta(\lambda-k)$. We only show this in detail for completeness; one would not follow such a convoluted path to arrive back at an exponential discount but this approach holds for richer priors. The derivations can be found in the Appendix F.

**Three cases:**

1. **Delta hazard prior**: $p(\lambda) = \delta(\lambda-k)$

2. **Exponential hazard prior**: $p(\lambda) = \frac{1}{k}e^{-\lambda/k}$

3. **Uniform hazard prior**: $p(\lambda) = \frac{1}{k}$ for $\lambda \in [0,k]$

For the three cases we begin with the Laplace transform on the prior $p(\lambda) = \int_{\lambda=0}^{\infty} p(\lambda)e^{-\lambda t}d\lambda$ and then chnage the variables according to the relation between $\gamma = e^{-\lambda}$, Equation 17.

## F.1 DELTA HAZARD PRIOR

A delta prior $p(\lambda) = \delta(\lambda - k)$ on the hazard rate is consistent with exponential discounting.

$$\int_{\lambda=0}^{\infty} p(\lambda)e^{-\lambda t}d\lambda = \int_{\lambda=0}^{\infty} \delta(\lambda - k)e^{-\lambda t}d\lambda$$
$$= e^{-kt}$$

where $\delta(\lambda - k)$ is a Dirac delta function defined over variable $\lambda$ with value $k$. The change of variable $\gamma = e^{-\lambda}$ (equivalently $\lambda = -\ln\gamma$) yields differentials $d\lambda = -\frac{1}{\gamma}d\gamma$ and the limits $\lambda = 0 \to \gamma = 1$ and $\lambda = \infty \to \gamma = 0$. Additionally, the hazard rate value $\lambda = k$ is equivalent to the $\gamma = e^{-k}$.

$$d(t) = \int_{\lambda=0}^{\infty} p(\lambda)e^{-\lambda t}d\lambda$$
$$= \int_{\gamma=1}^{0} \delta(-\ln\gamma - k)\gamma^t \left(-\frac{1}{\gamma}d\gamma\right)$$
$$= \int_{\gamma=0}^{1} \delta(-\ln\gamma - k)\gamma^{t-1}d\gamma$$
$$= e^{-kt}$$
$$= \gamma_k^t$$

where we define a $\gamma_k = e^{-k}$ to make the connection to standard RL discounting explicit. Additionally and reiterating, the use of a single discount factor, in this case $\gamma_k$, is equivalent to the prior that a *single* hazard exists in the environment.

## F.2 EXPONENTIAL HAZARD PRIOR

Again, the change of variable $\gamma = e^{-\lambda}$ yields differentials $d\lambda = -\frac{1}{\gamma}d\gamma$ and the limits $\lambda = 0 \to \gamma = 1$ and $\lambda = \infty \to \gamma = 0$.

$$\int_{\lambda=0}^{\infty} p(\lambda)e^{-\lambda t}d\lambda = \int_{\gamma=1}^{0} p(-\ln\gamma)\gamma^t \left(-\frac{1}{\gamma}d\gamma\right)$$
$$= \int_{\gamma=0}^{1} p(-\ln\gamma)\gamma^{t-1}d\gamma$$

where $p(\cdot)$ is the prior. With the exponential prior $p(\lambda) = \frac{1}{k}\exp(-\lambda/k)$ and by substituting $\lambda = -\ln\gamma$ we verify Equation 9

$$\int_0^1 \frac{1}{k}\exp(\ln\gamma/k)\gamma^{t-1}d\gamma = \frac{1}{k}\int_0^1 \exp(\ln\gamma^{1/k})\gamma^{t-1}d\gamma$$
$$= \frac{1}{k}\int_0^1 \gamma^{1/k+t-1}d\gamma$$
$$= \frac{1}{k}\frac{1}{\frac{1}{k}+t}\gamma^{1/k+t}\Big|_{\gamma=0}^{1}$$
$$= \frac{1}{1+kt}$$

### F.3 UNIFORM HAZARD PRIOR

Finally if we hold a uniform prior over hazard, $\frac{1}{k}$ for $\lambda \in [0, k]$ then Sozou (1998) shows the Laplace transform yields

$$
\begin{aligned}
d(t) &= \int_0^\infty p(\lambda)e^{-\lambda t}d\lambda \\
&= \frac{1}{k}\int_0^k e^{-\lambda t}d\lambda \\
&= -\frac{1}{kt}e^{-\lambda t}\Big|_{\lambda=0}^k \\
&= \frac{1}{kt}\left(1 - e^{-kt}\right)
\end{aligned}
$$

Use the same change of variables to relate this to $\gamma$. The bounds of the integral become $\lambda = 0 \to \gamma = 1$ and $\lambda = k \to \gamma = e^{-k}$.

$$
\begin{aligned}
d(t) &= -\frac{1}{k}\int_{\gamma=1}^{e^{-k}} \gamma^{t-1}d\gamma \\
&= \frac{1}{kt}\gamma^t\Big|_{\gamma=e^{-k}}^1 \\
&= \frac{1}{kt}\left(1 - e^{-kt}\right)
\end{aligned}
$$

which recovers the discounting scheme.

## G   DETERMINING THE $\gamma$ INTERVAL

We provide further detail for which $\gamma$ we choose to model and motivation why. We choose a $\gamma_{\max}$ which is the largest $\gamma$ to learn through Bellman updates. If we are using $k$ as the hyperbolic coefficient in Equation 19 and we are approximating the integral with $n_\gamma$ our $\gamma_{\max}$ would be

$$
\gamma_{\max} = \left(1 - b^{n_\gamma}\right)^k \tag{27}
$$

However, allowing $\gamma_{\max} \to 1$ get arbitrarily close to 1 may result in learning instabilities Bertsekas (1995). Therefore we compute an exponentiation base of $b = \exp(\ln(1 - \gamma_{\max}^{1/k})/n_\gamma)$ which bounds our $\gamma_{\max}$ at a known stable value. This induces an approximation error which is described more in Appendix H.

## H   APPROXIMATION ERRORS

Instead of evaluating the upper bound of Equation 9 at 1 we evaluate at $\gamma_{\max}$ which yields $\gamma_{\max}^{kt}/(1+kt)$. Our approximation induces an error in the approximation of the hyperbolic discount.

This approximation error in the Riemann sum increases as the $\gamma_{\max}$ decreases as evidenced by Figure 12. When the maximum value of $\gamma_{\max} \to 1$ then the approximation becomes more accurate as supported in Table 5 up to small random errors.

## I   ESTIMATING HYPERBOLIC COEFFICIENTS

As discussed, we can estimate the hyperbolic discount in two different ways. We illustrate the resulting estimates here and resulting approximations. We use lower-bound Riemann sums in both cases for simplicity but more sophisticated integral estimates exist.

As noted earlier, we considered two different integrals for computed the hyperbolic coefficients. Under the form derived by the Laplace transform, the integrals are sharply peaked as $\gamma \to 1$. The difference in integrals is visually apparent comparing in Figure 13.

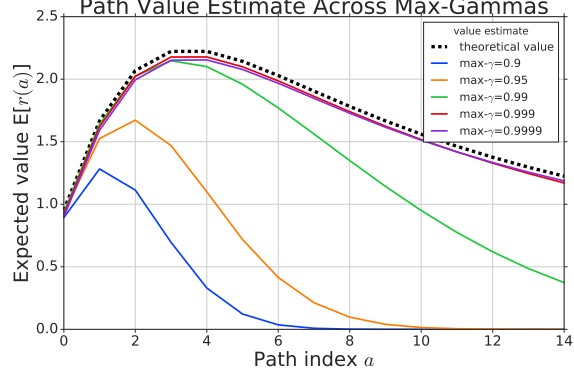

Figure 12: By instead evaluating our integral up to $\gamma_{\max}$ rather than to 1, we induce an approximation error which increases with $t$. Numerical results in Table 5.

| Discount function | MSE |
|---|---|
| max-$\gamma$=0.999 | **0.002** |
| max-$\gamma$=0.9999 | 0.003 |
| max-$\gamma$=0.99 | 0.233 |
| max-$\gamma$=0.95 | 1.638 |
| max-$\gamma$=0.9 | 2.281 |

Table 5: The average mean squared error (MSE) over each of the paths in Figure 12.

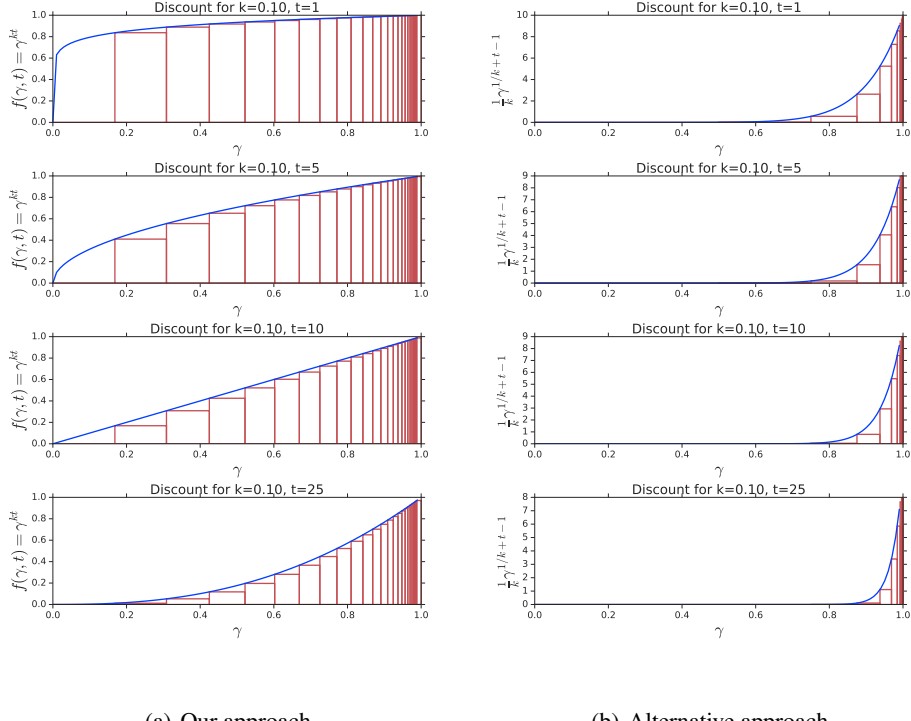

(a) Our approach.

(b) Alternative approach.

Figure 13: Comparison of hyperbolic coefficient integral estimation between the two approaches.
(a) We approximate the integral of the function $\gamma^{kt}$ via a lower estimate of rectangles at specific $\gamma$-values. The sum of these rectangles approximates the hyperbolic discounting scheme $1/(1 + kt)$ for time $t$.
(b) Alternative form for approximating hyperbolic coefficients which is sharply peaked as $\gamma \to 1$ which led to larger errors in estimation under our initial techniques.

## J  PERFORMANCE OF DIFFERENT REPLAY BUFFER PRIORITIZATION SCHEME

As found through our ablation study in Figure 7, the Multi-Rainbow auxiliary task interacted poorly with the prioritized replay buffer when the TD-errors were averaged evenly across all heads. As an alternative scheme, we considered prioritizing according to the largest $\gamma$, which is also the $\gamma$ defining the $Q$-values by which the agent acts.

The (preliminary[5]) results of this new prioritization scheme is in Figure 14.

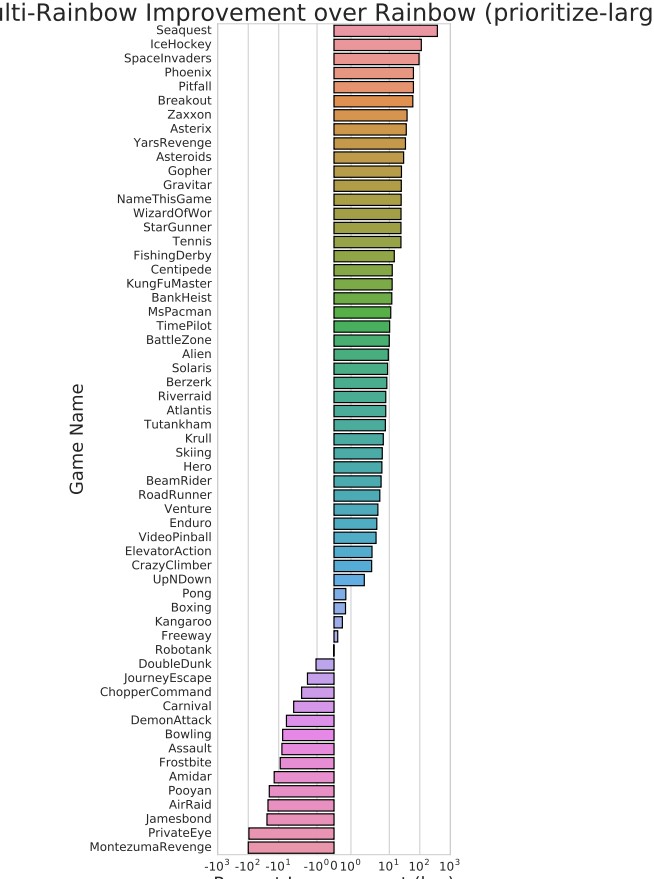

Figure 14: The (preliminary) performance improvement over Rainbow using the multi-horizon auxiliary task in Atari Learning Environment when we instead prioritize according to the TD-errors computed from the largest $\gamma$ (3 seeds each).

To this point, there is evidence that prioritizing according to the TD-errors generated by the largest gamma is a better strategy than averaging.

---

[5]These runs have been computed over approximately 100 out of 200 iterations and will be updated for the final version.

## K   HYPERPARAMETERS

For all our experiments in DQN Mnih et al. (2015), C51 Bellemare et al. (2017) and Rainbow Hessel et al. (2018), we benchmark against the baselines set by Castro et al. (2018) and we use the default hyperparameters for each of the respective algorithms. That is, our Multi-agent uses the same optimization, learning rates, and hyperparameters as it's base class.

| Hyperparameter | Value |
|---|---|
| Runner.sticky_actions | Sticky actions prob 0.25 |
| Runner.num_iterations | 200 |
| Runner.training_steps | 250000 |
| Runner.evaluation_steps | 125000 |
| Runner.max_steps_per_episode | 27000 |
| | |
| WrappedPrioritizedReplayBuffer.replay_capacity | 1000000 |
| WrappedPrioritizedReplayBuffer.batch_size | 32 |
| | |
| RainbowAgent.num_atoms | 51 |
| RainbowAgent.vmax | 10. |
| RainbowAgent.update_horizon | 3 |
| RainbowAgent.min_replay_history | 20000 |
| RainbowAgent.update_period | 4 |
| RainbowAgent.target_update_period | 8000 |
| RainbowAgent.epsilon_train | 0.01 |
| RainbowAgent.epsilon_eval | 0.001 |
| RainbowAgent.epsilon_decay_period | 250000 |
| RainbowAgent.replay_scheme | 'prioritized' |
| RainbowAgent.tf_device | '/gpu:0' |
| RainbowAgent.optimizer | @tf.train.AdamOptimizer() |
| | |
| tf.train.AdamOptimizer.learning_rate | 0.0000625 |
| tf.train.AdamOptimizer.epsilon | 0.00015 |
| | |
| HyperRainbowAgent.number_of_gamma | 10 |
| HyperRainbowAgent.gamma_max | 0.99 |
| HyperRainbowAgent.hyp_exponent | 0.01 |
| HyperRainbowAgent.acting_policy | 'largest_gamma' |

Table 6: Configurations for the Multi-C51 and Multi-Rainbow used with Dopamine Castro et al. (2018).

## L   AUXILIARY TASK RESULTS

Final results of the multi-horizon auxiliary task on Rainbow (Multi-Rainbow) in Table 7.

| Game Name | DQN | C51 | Rainbow | Multi-Rainbow |
|---|---|---|---|---|
| AirRaid | 8190.3 | 9191.2 | **16941.2** | 12659.5 |
| Alien | 2666.0 | 2611.4 | 3858.9 | **3917.2** |
| Amidar | 1306.0 | 1488.2 | **2805.7** | 2477.0 |
| Assault | 1661.6 | 2079.0 | **3815.9** | 3415.1 |
| Asterix | 3772.5 | 15289.5 | 19789.2 | **24385.6** |
| Asteroids | 844.7 | 1241.5 | 1524.1 | **1654.5** |
| Atlantis | **935784.0** | 894862.0 | 890592.0 | 923276.7 |
| BankHeist | 723.5 | 863.4 | **1209.0** | 1132.0 |
| BattleZone | 20508.5 | 28323.2 | **42911.1** | 38827.1 |
| BeamRider | 6326.4 | 6070.6 | 7026.7 | **7610.9** |
| Berzerk | 590.3 | 538.3 | 864.0 | **879.1** |
| Bowling | 40.3 | 49.8 | **68.8** | 62.9 |
| Boxing | 83.3 | 83.5 | 98.8 | **99.3** |
| Breakout | 146.6 | **254.1** | 123.9 | 162.5 |
| Carnival | 4967.9 | 4917.1 | **5211.8** | 5072.2 |
| Centipede | 3419.9 | **8068.9** | 6878.0 | 6946.6 |
| ChopperCommand | 3084.5 | 6230.4 | 13415.1 | **13942.9** |
| CrazyClimber | 113992.2 | 146072.3 | 151454.9 | **160161.0** |
| DemonAttack | 7229.2 | 8485.1 | **19738.0** | 14780.9 |
| DoubleDunk | -4.5 | 2.7 | **22.6** | 21.9 |
| ElevatorAction | 2434.3 | 73416.0 | 81958.0 | **85633.3** |
| Enduro | 895.0 | 1652.9 | 2290.1 | **2337.5** |
| FishingDerby | 12.4 | 16.6 | 44.5 | **45.1** |
| Freeway | 26.3 | **33.8** | 33.8 | 33.8 |
| Frostbite | 1609.6 | 4522.8 | **8988.5** | 7929.7 |
| Gopher | 6685.8 | 8301.1 | 11749.6 | **13664.6** |
| Gravitar | 339.1 | 709.8 | 1293.0 | **1638.7** |
| Hero | 17548.5 | 34117.8 | 47545.4 | **50141.8** |
| IceHockey | -5.0 | -3.3 | 2.6 | **6.3** |
| Jamesbond | 618.3 | 816.5 | **1263.8** | 773.4 |
| JourneyEscape | -2604.2 | -1759.1 | **-818.1** | -1002.9 |
| Kangaroo | 13118.1 | 9419.7 | 13794.0 | **13930.6** |
| Krull | 6558.0 | **7232.3** | 6292.5 | 6645.7 |
| KungFuMaster | 26161.2 | 27089.5 | 30169.6 | **31635.2** |
| MontezumaRevenge | 2.6 | **1087.5** | 501.3 | 800.3 |
| MsPacman | 3664.0 | 3986.2 | 4254.2 | **4707.3** |
| NameThisGame | 7808.1 | **12934.0** | 9658.9 | 11045.9 |
| Phoenix | 5893.4 | 6577.3 | 8979.0 | **23720.3** |
| Pitfall | -11.8 | -5.3 | **0.0** | 0.0 |
| Pong | 17.4 | 19.7 | 20.3 | **20.6** |
| Pooyan | 3800.8 | 3771.2 | **6347.7** | 4670.0 |
| PrivateEye | 2051.8 | 19868.5 | **21591.4** | 888.9 |
| Qbert | 11011.4 | 11616.6 | 19733.2 | **20817.4** |
| Riverraid | 12502.4 | 13780.4 | **21624.2** | 21421.2 |
| RoadRunner | 40903.3 | 49039.8 | **56527.4** | 55613.0 |
| Robotank | 62.5 | 64.7 | **67.9** | 67.2 |
| Seaquest | 2512.4 | 38242.7 | 11791.5 | **64985.0** |
| Skiing | **-15314.9** | -17996.7 | -17792.9 | -15603.3 |
| Solaris | 2062.7 | 2788.0 | 3061.9 | **3139.9** |
| SpaceInvaders | 1976.0 | 4781.9 | 4927.9 | **8802.1** |
| StarGunner | 47174.3 | 35812.4 | 58630.5 | **72943.2** |
| Tennis | -0.0 | **22.2** | 0.0 | 0.0 |
| TimePilot | 3862.5 | 8562.7 | 12486.1 | **14421.7** |
| Tutankham | 141.1 | 253.1 | 255.6 | **264.9** |
| UpNDown | 10977.6 | 9844.8 | 42572.5 | **50862.3** |
| Venture | 88.0 | 1430.7 | 1612.4 | **1639.9** |
| VideoPinball | 222710.4 | 594468.5 | **651413.1** | 650701.1 |
| WizardOfWor | 3150.8 | 3633.8 | 8992.3 | **9318.9** |
| YarsRevenge | 25372.0 | 12534.2 | 47183.8 | **49929.4** |
| Zaxxon | 5199.9 | 7509.8 | 15906.2 | **21921.3** |

Table 7: Multi-Rainbow agent returns versus the DQN, C51 and Rainbow agents of Dopamine Castro et al. (2018).

