# OpenReview forum: "Hyperbolic Discounting and Learning Over Multiple Horizons"
_ICLR.cc/2020/Conference — Reject_

### Official Review · AnonReviewer3 · 2019-10-22
**Official Blind Review #2**

**Rating:** 6

**Review:**

The paper studied the discounting factor of RL under the uncertain hazard and non-trivial intertemporal decisions, by proposing a hyperbolic discounting strategy. The authors show the equivalence between the hyperbolic discount function and the temporal difference learning techniques in RL. Experimental results show the performance of the proposed method by comparing it with state-of-the-art. In general,  the text is well written and easy to follow. However,  I am not familiar with the context of hyperbolic computation.  I do not know any related works or what to expect from the results. I could not find anything wrong with this paper, but also do not have any intelligent questions to ask. Therefore, I am not sure about the technical contribution of this paper to the area.


**Experience Assessment:**

I do not know much about this area.

**Review Assessment: Checking Correctness Of Derivations And Theory:**

I carefully checked the derivations and theory.

**Review Assessment: Checking Correctness Of Experiments:**

I carefully checked the experiments.

**Review Assessment: Thoroughness In Paper Reading:**

I read the paper at least twice and used my best judgement in assessing the paper.

---

> ### Author Response · Authors · 2019-11-08
> **Further Context and Contributions**
>
> Thanks for your review!
>
> To provide more context, reinforcement learning almost exclusively considers exponentially discounted models.  As noted in our work, the closest related work is that of Kurth-Nelson & Redish (2009), which demonstrated the plausibility of using many exponentially-discounted micro agents to approximate hyperbolic discounting in tabular environments.  However, this is, to the best of our knowledge, the first work demonstrating non-exponential discount functions in deep RL.  Additionally, we show that our new model structure can be used to model more than hyperbolic discount functions and can be extended to other non-hyperbolic discount functions.
>
> Additionally, we would also highlight to the reviewer the importance of the multi-horizon auxiliary task.  By including this new form of auxiliary task, we improve over a state-of-the-art algorithm, Rainbow.  This is particularly interesting because Rainbow is a composite of various RL algorithmic improvements and we show that our new task is additionally beneficial on top of these other strategies.
>
> Please tell us how we can further clarify questions you have or elaborate on details of the technical contributions.

---

### Official Review · AnonReviewer1 · 2019-10-24
**Official Blind Review #1**

**Rating:** 3

**Review:**

This paper argues that hyperbolic and other non-exponential discounting mechanisms have been more utilized by humans and animals for value preferences than exponential discounting as widely used in RL literature. The authors claim that hyperbolic discounting mechanisms are especially preferred in the setting of maintaining uncertainty over the prior belief of the hazard rate in the environment and propose an efficient approximation of the Q function with hyperbolic and other non-exponential discounting mechanisms as a weighted sum of Q-functions with the standard exponential discounting factor. The paper shows empirical evidence that hyperbolic discounting function can more accurately estimate the value in a vanilla Pathworld environment and also demonstrate that the approximated multi-horizon Q functions can improve performance on ALE, which is largely attributed to learning over multi-horizons as an auxiliary task.

Overall, this paper is an extension of the prior work Kurth-Nelson & Redish (2009). It seems to me that the difference between this paper and Kurth-Nelson & Redish (2009) is that in this paper, the approximated Q-value with hyperbolic discounting function is a weighted sum over each Q-values using exponential discounting factor gamma, while in Kurth-Nelson & Redish (2009), the Q-value is estimated by sampling one Q-value based on the distribution of the gamma. Another difference as claimed in the paper is that the authors also present an approximation of other non-hyperbolic discounting functions. Those extensions seem a bit incremental and are not well supported by the experimental results in the paper. The authors didn’t compare their method to Kurth-Nelson & Redish (2009) in both Pathworld and Atari 2600, which seems insufficient to demonstrate the point that the extension is useful. Moreover, the authors didn’t conduct experiments on non-hyperbolic discounting functions, which makes the claim that approximating non-hyperbolic discounting functions is one of the paper’s contributions unsubstantiated empirically.

In Section 6, the experiments in Atari 2600 show that Hyper-Rainbow and Multi-Rainbow have similar performance, and the authors claim that such results imply that learning over multi-horizons as an auxiliary task is more useful than choosing different discounting schemes. This seems to contradict the whole point of the paper that the hyperbolic discounting mechanism is biologically more reasonable than exponential discounting functions. The authors then claim that learning Q-values with multiple discounting factors is also one of the paper’s contributions, which makes the paper look like a list of tricks to get RL to work in the multi-task setting rather than a unifying framework of the hyperbolic discounting scheme. I think the authors might need to pick a deep RL domain similar to Pathworld, i.e. an environment with risk increasing as the length of the trajectory increases, but in high-dimensional state space and continuous action space, to better demonstrate the effectiveness of the method.

**Experience Assessment:**

I have published one or two papers in this area.

**Review Assessment: Checking Correctness Of Derivations And Theory:**

I assessed the sensibility of the derivations and theory.

**Review Assessment: Checking Correctness Of Experiments:**

I assessed the sensibility of the experiments.

**Review Assessment: Thoroughness In Paper Reading:**

I read the paper at least twice and used my best judgement in assessing the paper.

---

> ### Author Response · Authors · 2019-11-08
> **Response and Paths to Improvement**
>
> Thanks for the help improving our paper!
>
> We believe that some of the criticisms are addressed when presented in a new light:  we propose learning over multiple horizons as a useful auxiliary task in deep RL and which allows for approximating non-exponential discount schemes.
>
> It was not our intention to imply that hyperbolic discounting should _always_ be used and we aimed through our ablations in Atari 2600 to show cases where this is not helpful.  Please tell us if you agree/disagree with below proposals and we’ll iterate to improve them.
>
> *Critique*:  [Limited differences between [1]]
> *Response*:  We believe the algorithmic contributions beyond [1] are more substantial.  Their paper investigates learning completely separated tabular Q-values in far simpler environments.  For instance, as an environment example, their work considers an MDP with only five states and two actions.  In contrast, Atari 2600's state space is high dimensional and extends across many different types of tasks.  In order to do this, we constructed a new architecture that has a shared convolutional body and models each time horizon separately from a shared representation.  Furthermore, we extend beyond their special-case of hyperbolic discounting and show that this can be more broadly repurposed, in-line with new hazard priors.  One might claim that designing deep-RL algorithms based on older algorithms is not a meaningful contribution, but that would invalidate much of the work in the field (DQN, actor-critic architectures, Feudal Networks...).  We will further distinguish our work in the final version.
>
> *Critique*: [Baseline to [1]]
> *Response*:  This is an underspecified baseline as stated, but perhaps we can work with you here to devise a reasonable one.  [1] use completely separated tabular models to produce the hyperbolic value.  For Atari 2600, are you suggesting that we run an ensemble of deep Q-networks, each with their own discount factor?  We can run that if so, however, given the results of the multi-horizon auxiliary task, we expect our model will perform considerably better than this.
>
> *Critique*: [Evaluate non-hyperbolic discount functions]
> *Response*:  We will add non-hyperbolic discount functions and show our method working there.  We initially had these but removed them due to length constraints.  Our findings are clear - our method is able to approximate other non-hyperbolic discount functions expressible from our d-discounted criterion (Equation 4) and we will include that for completeness.
>
> *Critique*: “...the authors claim that such results imply that learning over multi-horizons as an auxiliary task is more useful than choosing different discounting schemes. This seems to contradict the whole point of the paper that the hyperbolic discounting mechanism is biologically more reasonable than exponential discounting functions.”
> *Response*:  Our intention with this work was not to make a preferential statement about one discounting scheme over another.  Rather, our objective was to note that the deep RL community makes this discounting decision implicitly with little deliberation and show generalizations.  This work provides tools to model different discount functions while still using familiar frameworks in an efficient manner.  Therefore instead of contradicting our paper, we believe that our ablation study and discovery of the multi-horizon auxiliary task represented a sound scientific process and strongly augments the paper.  A weaker scientific inquiry would failed to run this ablation and presented strong hyperbolic results in isolation (both hyperbolic and multi-horizon agents exceed Rainbow).  We can rewrite the paper for the final version with this intention more clearly stated to reduce confusion to readers if you believe this would be beneficial.
>
> *Critique*:  [Multi-horizon learning is an RL trick]
> *Response*:  We believe that modeling the environment on many time scales is actually more fundamental and not a RL trick (of course, one person’s algorithm is another’s trick).  To elaborate, when a MDP is modeled with a single time scale, there can be ambiguity about the future reward sequence given the scalar discounted value.  Different reward sequences may map to the same value.  For example, $V(s) = 1$ might mean $r_0 = 1$ and $r_t=0 \forall t>0$.  Alternatively, $V(s) = 1$ might mean $r_9=10$ and $r_t=0$ o/w with a $\gamma=0.775$ (because $0.775^9 = 0.1$).  However, by modeling the world on multiple time scales it _distinguishes_ these two different states!
>
> *Critique*: [New deep RL env. beneficial]
> *Response*:  We agree that this would be useful, but have failed to find RL environments with such properties.  Most common RL environments are repeatable and don’t have dynamic hazard structure.  We can augment the ALE to have a dynamic hazard and present that here if the reviewer will think this further substantiates the method. Thoughts?
>
>
> [1]  Kurth-Nelson & Redish (2009)

---

### Official Review · AnonReviewer4 · 2019-11-09
**Official Blind Review #4**

**Rating:** 6

**Review:**

The paper investigates hyperbolic discounting as a more biologically plausible alternative to exponential discounting in reinforcement learning. First, it formulates a notion of hazard in MDPs as constant exponential discounting and shows that hyperbolic discounting is consistent with uncertainty over the hazard rate. The paper then shows how value functions learned with exponential discounting can be used to approximate value functions with other forms of discounting. Specifically, the paper shows in section 4 how exponentially-discounted value functions can be used to approximate hyperbolically discounted value functions. The paper then presents experiments on a small MDP and Atari 2600 games, showing that learning discounted action values with many different discount rates as an auxiliary task improves performance on most Atari games.

Overall, I very slightly tend towards accepting this paper for publication. While the idea of hyperbolic discounting didn't seem to pay off in terms of performance on the Atari 2600 games, the idea still has some merit as being more biologically plausible than exponential discounting, and may perform better on a more suitable environment. In addition, the discovery that learning many action-value functions with different exponential discounting rates as auxiliary tasks improves performance is quite interesting.

Notes:
- The footnote on the first page seems to just trail off. It would be better to explicitly state what the takeaway is. I'm guessing humans and animals in general would prefer \$1M now in the first scenario, but \$1.1M in the second scenario?
- The idea of 1/(1-gamma) being an "effective horizon" seems questionable. For gamma=0.9 the "effective horizon" (aka expected number of timesteps before termination) would be 1/(1-.9)=10 timesteps. However, 0.9^(10)=~0.3486 means ~34% of the probability density is still to the right of that number, so calling 1/(1-gamma) an "effective horizon" seems to violate what we normally mean by "horizon".
- A single hazard rate for the entire environment seems counterintuitive. Some states in life are definitely more hazardous than others, which suggests state-dependent discounting might be an interesting idea to explore.
- Many people in the field think we shouldn't even be doing discounting at all (section 10.4 of Reinforcement Learning by Sutton and Barto), only total reward for episodic problems and average reward for continuing problems.
- Pathworld experiments seem like they don't show anything useful; the hyperbolic discounting matches the true hazard behaviour of the environment (i.e., the hazard rate is drawn from a distribution) so it performs better than exponential discounting. However, if the environment had a fixed hazard rate then exponential discounting would perform better.
- Using only 3 random seeds does not seem like enough for the experiments. Also there is no measure of standard error or statistical significance on the graphs.
- Introducing a hyperparameter that increases computation and memory does not really seem like it's solving the problem, and certainly not "efficiently" as claimed in the abstract.
- How was the value of the new hyperparameter set for the experiments? Were several values tried? More importantly, how can a good value for the new hyperparameter be chosen without prior knowledge of the environment?
- Atari 2600 games don't really seem like a great environment to showcase the benefits of hyperbolic discounting.
- It would be interesting to see a comparison with existing auxiliary task approaches, but due to space constraints should probably be left for future work.

**Experience Assessment:**

I have read many papers in this area.

**Review Assessment: Checking Correctness Of Derivations And Theory:**

I did not assess the derivations or theory.

**Review Assessment: Checking Correctness Of Experiments:**

I assessed the sensibility of the experiments.

**Review Assessment: Thoroughness In Paper Reading:**

I read the paper at least twice and used my best judgement in assessing the paper.

---

> ### Author Response · Authors · 2019-11-13
> **Thanks for Your Comments - Next Steps**
>
> Thanks for reading our work and your help in improving our paper!
>
> *Critique* [Motivating footnote for hyperbolic discounting is unclear]
> *Response*:  We will change the footnote for our camera ready version to include the explicit conclusion that the reviewer correctly conjectured.
>
> *Critique* [Effective horizon is not a good term]
> *Response* Yes, we will clarify that this is not a hard-horizon, but rather, it implies that approximately 1 - 1/e of the probability mass lies ahead of that horizon.
>
> *Critique* [A single hazard rate is counterintuitive]
> *Response* We agree! We discuss state-action dependent discounting (Pitis, 2019) as an interesting avenue for future research.  Also, to reiterate and clarify, the choice of a single hazard rate is equivalent to choosing a single discount factor gamma - the canonical approach in RL.  Beyond a single value, we would also agree that a fixed hazard rate _distribution_ is also too restrictive because this distribution also changes with state-action tuples.  We save these extensions for future work.
>
> *Critique* [Sutton suggests that no discounting may be needed]
> *Response*  Yes, we are aware of this result in an infinite horizon setting, however, we note that Section 10.4 also leaves open the case that discounting is still relevant in the episodic case:  “The discounted case is still pertinent, or at least possible, for the episodic case.” - Sutton & Barto (2017).  That said, we are not trying to take a stance on whether or not to discount in episodic settings, but would like to bring a new understanding and tools to bear if one chooses to discount.
>
> *Critique* [Reason for Pathworld Experiments]
> *Response* We choose to use Pathworld with three purposes in mind (1) to give a concrete example to the reader of a foraging-like environment where alternative discount schemes are beneficial (2) as a validation that our approach is working and (3) to show that maintaining uncertainty over the hazard rate/discount rate is better matches value functions even if the prior coefficient was wrong (Figure 9; Table 2) or if the functional form was wrong (Figure 10; Table 3).
>
> *Critique*  [Three seeds insufficient]
> *Response*  We agree more seeds is almost always desirable but we encourage the reviewer and readers not to focus on the performance benefits of any particular game, but rather, the improvement over all games.  To elaborate, we consider the entire ALE suite of ~60 games.  We use 3 seeds for our model (yielding 180 measures) and 5 seeds for the baseline (300 measures).  Being subject to computational constraints (each seed required a P100 GPU for 1 week), we focused on measuring the generalization of our method over the ALE environment rather than a precise measure on fewer environments.   For instance, an alternative and very wasteful approach would be to use 180 seeds in the game of Pong and have a _highly_ precise measure of improvements in that single game, but would have no information about generalization beyond.  But if the reviewer still believes 180 measures is insufficient and recommends a more accurate _per game_ measure for the camera ready, we can launch an additional 2-3 seeds resulting in 300 to 360 measures.  Recommendation?
>
> *Critique*  [This method is not efficient]
> *Response*  We agree that this method is doing more computation, however, it is efficient in the sense that it builds off a shared representation of the environment and the per-discount factor computations can be batched on modern accelerators.
>
> *Question*  [Hyperparameter choice?]
> *Response*  We sought an algorithm that was hyperparameter robust, therefore, we did not change any of the baseline Rainbow hyperparameters (lr, eps-decay schedule, optimizer, base convolutional architecture, target network update period...).  We only considered two hyperparameters:  `hyp_exponent=0.01` (the $k$ of $\frac{1}{1+kt}$ and the `number_of_gamma=10`.  A larger number of heads allows for better approximation of the non-exponential discount functions and the hyp_exponent ($k$) has a similar role as $\gamma$ in RL  e.g. if discounting hyperbolically, $d(t) = \frac{1}{1+kt}$, $k=0.01$ yields a discount rate of 0.5 at $t=100$ whereas a commonly used $\gamma=0.99$ yields a discount rate of 0.37 at $t=100$.  We used a random subset of 10 ALE games for our hyperparameter tuning but found the algorithm robust to number of heads but dependent on a reasonable $k$ parameter (match the time-scale of the ALE already shown by many works to be reasonable).
>
> *Critique*  [Atari 2600 not a good environment to showcase hyperbolic benefits]
> *Response*  We agree, but unfortunately many common RL environments lack the dynamic hazard properties necessary to see these benefits. We believe as future work that multi-agent RL may be an interesting avenue to investigate the benefits of non-exponential discounting.
>
> *Note*  [Comparison to other auxiliary tasks]
> *Response*  We agree but save it for future work.

---

> > ### Comment · AnonReviewer4 · 2019-11-14
> > **Responding to author's comments**
> >
> > Thank you for taking the time to read my comments and reply.
> >
> > I'd like to respond to a few of your replies:
> >
> > Random seeds: Please do not run more random seeds if they take 1 week on a P100 GPU. That seems incredibly wasteful of electricity and strikes me as a strong argument against using the ALE as a testing environment, especially when it's not a great fit for the research question being investigated.
> >
> > Hyperparameters: The fact that the extra hyperparameter (number of heads) doesn't require domain knowledge to set is really nice.
> > However, it would've been nice to see how increasing/decreasing the number of heads changes the performance and computational cost tradeoff. The fact that each random seed takes so long to run makes it infeasible to conduct such a hyperparameter study. As a result, we're left with a worse understanding than if the proposed method were run on a smaller environment where a more extensive empirical investigation would've been feasible. However, it's probably too late to change environments now. The results shown on the ALE are a good first step to a more in-depth investigation.

---

> > > ### Author Response · Authors · 2019-11-14
> > > **Discussion Response**
> > >
> > > No problem and we appreciate the active discussion!
> > >
> > > Agreed on the value of additional seeds here - we will not run more.  Yes, the relative insensitivity to number of heads is a nice feature.  And as you said, unfortunately doing this hyperparameter sweep in the ALE is computationally infeasible, however, we could run this on the Pathworld environment (at a tiny fraction of the computational cost).  Do you think that would be valuable to include for the final version?  If so, we'll add a new section in the Appendix and run these.

---

> > > > ### Comment · AnonReviewer4 · 2019-11-14
> > > > **Response**
> > > >
> > > > I would say don't bother. It could be interesting to see how the quality of approximation changes with different numbers of heads and could give a basis for recommending a good hyperparameter setting, but I don't think it's required, especially because the experiments found that hyperbolic discounting is not actually responsible for the performance improvement when using neural nets (and therefore the quality of approximation is not really important in the final method).
> > > >
> > > > It would be more interesting to see how the number of heads affects performance as auxiliary tasks with neural nets in follow-up work.

---

> > > > > ### Author Response · Authors · 2019-11-15
> > > > > **Auxiliary Task Future Work**
> > > > >
> > > > > OK - makes sense.  Yes, a deeper analysis of the auxiliary task, including as you suggested earlier, (1) measuring interaction with other auxiliary tasks and (2) performance as a function of number of heads would be useful future work.
> > > > >
> > > > > Finally, in a related note on our auxiliary task, we are using shallow function approximation built off the shared convolutional body. However, it is plausible that this design is not sufficiently expressive to model value functions at multiple time scales.  This was the first design we considered and it worked immediately so we never investigated it further, but deeper architectures may provide an additional benefit.

---

### Decision · Program_Chairs · 2019-12-19

**Decision:**

Reject

**Comment:**

While there was some support for the ideas presented in this paper, it was on the borderline, and ultimately did not make the cut for publication at ICLR.

Concerns were raised as to the significance of the contribution, beyond that of past work.